# Authentic Leadership of Nurses in the Hospital: A Scoping Review

**DOI:** 10.3390/healthcare13212713

**Published:** 2025-10-27

**Authors:** Ana Rita Santos, Patrícia Costa, Ana Rita Figueiredo, Elisabete Nunes, Paulo Cruchinho, Marta Santos, Pedro Lucas

**Affiliations:** 1Nursing Research Innovation and Development Centre of Lisbon (CIDNUR), Nursing School of Lisbon, 1600-190 Lisbon, Portugal; ana-rita-santos@campus.esel.pt (A.R.S.); patriciacosta@esel.pt (P.C.); enunes@esel.pt (E.N.); pjcruchinho@esel.pt (P.C.); prlucas@esel.pt (P.L.); 2Department Cardiac Surgery, Unidade Local de São José/Hospital de Santa Marta, R. de Santa Marta 50, 1169-024 Lisbon, Portugal; 3Nursing Administration Department, Nursing School of Lisbon, Avenida Prof. Egas Moniz, 1600-190 Lisbon, Portugal; 4Department of General Surgery and Gastroenterology, Instituto Português de Oncologia de Lisboa, 1099-023 Lisbon, Portugal; 5Escola Superior de Educação de Lisboa, Instituto Politécnico de Lisboa, Campus de Benfica do IPL, 1549-003 Lisbon, Portugal; martarpsantos06@gmail.com

**Keywords:** authentic leadership, hospital, nursing, nursing administration research, review

## Abstract

Background: The new challenges facing the healthcare sector require nursing leaders who are prepared to respond to patients’ healthcare needs and keep professionals motivated and satisfied while meeting the organisation’s objectives. To overcome these challenges, interest has grown in positive leadership styles, which in turn has sparked interest in authentic leadership. This line of thought on leadership has been predictive of various positive outcomes in the hospital context. Objective: Examine the characteristics of authentic leadership among nurses in the hospital context based on scientific evidence. Methods: Scoping review according to the Joanna Briggs Institute, consisting of three stages of research. First, a search was conducted in MEDLINE Ultimate, CINHAL Ultimate and MedicLatina through the EBSCOHost platform, LILACS and RCAAP for grey literature. Then, using the same keywords, we conducted a search in Scopus and ScienceDirect. Both searches covered the period between 2019 and 2024. Studies were included if they focused on authentic leadership in nurses working in hospital contexts; non-empirical studies, reviews, and those outside the nursing field were excluded. Results: Sixteen articles were included, 13 of which were cross-sectional, descriptive and/or correlational, 2 were quasi-experimental and 1 employed an exploratory and confirmatory factor analysis approach. The Authentic Leadership Questionnaire was used in a total of 11 articles. The studies showed that authentic leadership by nurses in a hospital context is significantly associated with relevant organisational and individual variables, such as team performance, organisational commitment, job satisfaction, reduced burnout, and the promotion of healthy work environments. The data also revealed discrepancies between the perceptions of leaders and their subordinates. Conclusion: Authentic leaders have a positive impact on the quality of nursing care, patient health outcomes, professional satisfaction and motivation to lead and the achievement of healthcare institution goals. Consequently, authentic leadership is a fundamental pillar for strengthening nursing teams, promoting trust, well-being and improvements in the quality of care. Nevertheless, the strength of the evidence is limited, given the predominance of cross-sectional studies and potential contextual biases, which should be considered when interpreting the findings.

## 1. Introduction

Leadership can be defined as a process in which an individual influences a group of individuals to achieve common goals [1,2]. Although leadership is considered to stem from personal experiences and contexts, it is acknowledged that it can be developed and/or improved [2,3,4]. Zhang et al. [5] point out that the leadership style adopted by the leader affects the attitudes and behaviours of employees.

The transformation of healthcare, with its increasing complexity, can jeopardise its humanisation [6], requiring competent, visionary, innovative, engaging, relational and caring leaders. Ultimately, it requires emotionally intelligent leaders who can respond to the people receiving care, the nursing team and the objectives of healthcare organisations [3]. The growing complexity of care poses new challenges, which can overshadow its humanisation [6].

There are several currents of thought in leadership, including authentic leadership. This has gained prominence as it has been shown to be a predictor of multiple positive outcomes in nursing [1]. The increase in research and interest in authentic leadership in recent years has been mainly motivated by the search for positive leadership [7]. Aly et al. [8] highlight that it is a positive leadership style that complements ethical and transformational leadership and has been shown to improve decision-making and enhance positive emotions and morale within the workforce. Its application in nursing is based on human values and care, which are at the heart of nursing practice [1,5,9].

Authentic leadership theory is defined as a pattern of transparent and ethical leadership behaviour that encourages openness in sharing information necessary for decision-making, while simultaneously accepting contributions from followers [10,11]. This pattern of leader behaviour is based on promoting positive psychological capabilities and a positive ethical climate [1,2,3].

Authentic leadership is a multidimensional concept characterized by self-awareness, relational transparency and internalized moral values. It emphasizes being true to oneself as a foundation for building trust, fostering collaboration and achieving positive outcomes in the workplace [12]. Its essence goes beyond the authenticity of the leader; it also involves the authenticity of the leader-follower relationship [13]. This focuses on making ethical decisions, allocating resources, sharing information and resolving clinical concerns, positively influencing the nursing practice environment, which ultimately influences the quality of care and patient safety [14].

Authentic leadership positively affects the three core components of healthcare services (patients, professionals and the work environment), contributes to the well-being of stakeholders, promotes healthy work environments and provides quality care [15]. The training of authentic leaders who have the skills to manage current challenges is essential [6] to ensure an effective response to health needs, obtain high-quality care, focus on people rather than tasks and achieve the desired outcomes. This current of leadership thought may be a long-term solution for nurse retention, since leaders act in accordance with justice and ethics and listen to team members, valuing them individually. The importance of leadership for the provision of high-quality care is unquestionable [6]. Nurse leaders should facilitate positive outcomes for patients, professional retention, nurse engagement in the organisation and the promotion of a healthy nursing practice environment [6,16]. Nurse leaders who embody the attributes of the conceptual framework of authentic leadership, specifically self-awareness, ethical and moral principles, relational wholeness, shared decision-making and care [6]. This can provide real support to nurses for a healthy professional life [6]. Wong and Laschinger [17] emphasise that authentic leaders within the nursing team, facilitate quality relationships. This, in turn, encourages the active involvement of staff in the workplace, ultimately contributing to greater satisfaction, productivity, and performance [17]. Giordano-Mulligan and Eckardt [6] reinforce that the authentic leadership theory suggests that the most authentic leaders are those who draw on life experiences, psychological capacities such as optimism, resilience and self-efficacy. These leaders maintain a solid moral and ethical perspective and build a supportive organisational climate to promote self-awareness and self-regulation [6]. Mondini et al. [2] also highlight that authentic leaders contribute to a positive organisational climate by valuing capabilities, promoting resilience and establishing trust, hope and optimism in the team. The theory further emphasises that the leader models and supports self-determination, promoting autonomy by empowering decision-making regarding work methods, procedures, pace and effort required to perform assigned tasks [17]. Batista et al. [18] reveal that leaders who demonstrate relational transparency, balanced processing, self-awareness and high ethical standards increase job satisfaction and performance among the nurses they lead. This relationship is confirmed by the results of the study by Lee et al. [12], which found a moderate and positive correlation between authentic leadership and a positive nursing practice environment and, conversely, a low and negative correlation between authentic leadership and burnout and intention to leave.

Authentic leadership plays an important role in emphasising the leader’s function as a facilitator of team development, creating a positive work environment with appropriate workloads, reducing emotional exhaustion and providing job satisfaction [18,19]. It should be noted that job satisfaction is essential for the functioning of the organisation, since greater satisfaction leads to lower turnover among nurses, which influences not only the sustainability of the workforce but also the financial management of institutions [18].

The application of authentic leadership in nursing is based on human values and care, which are at the core of nursing [1,5,9]. Given that the focus is on people and their relationships, it can be considered a theory that brings management closer to nursing, which humanises management. Authenticity in a leader influences people, teams, relationships and nursing practice environments, resulting in ultimate benefits for patients, professionals and healthcare institutions [3,14].

The nurse manager plays a key role in building a healthy and positive nursing practice environment and promoting quality care. Therefore, it is necessary to develop management skills to support the team and promote cohesion and collaborative work [20,21]. The nursing practice environment has an impact on nurses’ intention to leave, and management support and appreciation of their work increase professional satisfaction [21,22]. The improvement of nursing practice environment should be seen by nurse managers and hospital organisations as an investment that will be reflected in improved health outcomes for users, professional performance and quality of care, regardless of the context [21]. Dubois et al. [23] emphasise that providing excellent care requires not only investment in human resources but also the development of nursing practice environment, enabling nurses to perform with quality. A secure nursing practice environment is characterised by good professional relationships, management support, a balanced workload, autonomy, adequate resources and opportunities for career progression [20,21]. Its characteristics are modifiable and can be improved through administrative policies, namely through leadership training, promotion of continuity of care, and shared decision-making [14,24]. The American Association of Critical-Care Nurses (AACN) also recognises the role of work environments in nursing success and has defined, based on scientific evidence, six essential standards for building healthy environments, one of which is authentic leadership [25,26]. Authentic leadership has attracted more attention from researchers and health professionals over the past two decades [27]. The increase in research and interest in authentic leadership in recent years has been mainly motivated by the search for positive leadership [8,28,29]. It has gained prominence because it has been shown to be a predictor of multiple positive outcomes in nursing [1,3,30]. Thus, the adoption of authentic leadership by nursing leaders can positively transform healthcare organisations.

Malila et al. [31], in a scoping review of authentic leadership, identified four main themes: well-being at work, quality of patient care, work environment and promotion of authentic leadership. However, their review did not focus exclusively on nursing practice and did not incorporate the most recent scientific evidence. So, we decided to develop a scoping review in this field of knowledge to fill the time gap since the only known scoping review [31]. A literature review dated 2018 on authentic leadership was also found that does not specifically focus on nursing profession [27], but it was considered to define the time limit of the research strategy. Considering these gaps, we identified the need to update and narrow the scope of the available knowledge. Therefore, we developed this scoping review to synthesise the recent literature specifically on authentic leadership among nurses in hospital settings. This scoping review can contribute to identify gaps in knowledge and support the development of future systematic reviews and evidence-based interventions. Thus, the objective of this review was to examine, in the scientific evidence, the characteristics of authentic leadership of nurses in a hospital context. With these results, we aim to provide relevant contributions to nursing care, with the ultimate goal of improving its quality and safety.

## 2. Materials and Methods

In this scoping review we followed the methodology proposed by the Joanna Briggs Institute (JBI) for scoping reviews [32] and the recommendations of the Reporting Items for Systematic Reviews and Meta-Analyses Extension for Scoping Reviews (PRISMA ScR) [33]. The protocol was previously registered in the Open Science Framework (OSF) (https://osf.io/8edqn, accessed on 5 December 2024). We used meta-aggregation according to Lockwood et al. [34].

The review question was: “How is authentic leadership by nurses in a hospital context characterised in the scientific evidence?”. The review question was defined according to the PCC mnemonic, considering the Population (P)—nurses; the Concept (C)—authentic leadership; and the Context (C)—the hospital. The defined objective was examine, in the scientific evidence, the characteristics of authentic leadership of nurses in a hospital context.

### 2.1. Eligibility Criteria

The eligibility criteria were defined based on the PCC mnemonic (participants, concept, context), in accordance with the methodology proposed by the Joanna Briggs Institute (JBI) [33].

Population: Studies focusing on nurses were included, regardless of their professional category. Studies whose main target group was other health professionals, students or patients were excluded to ensure that the data analysed referred exclusively to the perception, performance or impact of authentic leadership in nursing.Concept: Studies that explicitly addressed the concept of authentic leadership were considered eligible. Studies that addressed other leadership styles or theories were excluded, except when directly compared with authentic leadership. Studies that applied validated instruments to measure authentic leadership or that addressed its effects on related variables were also included.Context: Only studies conducted in a hospital context were included, regardless of the type of institution. Studies conducted in other healthcare contexts, such as community, primary care, nursing homes or educational institutions, were excluded so that the focus on the organisational reality of hospitals is maintained.

### 2.2. Type of Sources

This scoping review covered a wide variety of sources, including primary studies with quantitative, qualitative and mixed-method approaches, as well as secondary studies, such as literature reviews and conceptual analyses. Theoretical and opinion articles, dissertations and master’s and doctoral theses, grey literature (unpublished studies), as well as books and book chapters relevant to the topic were also considered.

### 2.3. Search Strategy

The database search was conducted in September 2024. In order to locate articles relevant to this review, a search was conducted of published and unpublished documents, covering electronic databases and grey literature sources. The electronic databases consulted included:CINAHL Ultimate (by EBSCOHost);MEDLINE Ultimate (by EBSCOHost);MedicLatina (by EBSCOHost);Scopus;ScienceDirect.

For grey literature, searches were conducted in the following databases:LILACS;RCAAP.

The research strategy aimed to find studies available in the databases used on the selected topic that would answer the review question. The research strategy was carried out in three stages.

First stage: Keywords were identified using the Health Sciences Descriptors (DeCS) and Medical Subject Headings (Mesh). The descriptors used were “Authentic”, “Hospital Context”, “Leadership” and “Nurse”, combined with the Boolean operators AND and OR. Thus, the search performed using the descriptors to obtain the articles was: Authentic OR Leadership AND Nurse AND Hospital context.Second stage: Initially, research was conducted using the EBSCOHost platform, accessed through the virtual library of the Lisbon School of Nursing. This provided access to the CINAHL Ultimate, MEDLINE Ultimate and MedicLatina databases and the descriptors defined for the review were applied, with the search being carried out at the title and abstract level. The platforms available on the internet, such as Scopus and ScienceDirect, were also accessed, using the same descriptors.Third stage: Studies were identified in the bibliographic references of the included articles and others derived from expert recommendations.

This review was limited to studies published in English, Spanish, and Portuguese, as these are the languages mastered by the reviewers, allowing for accurate interpretation and analysis. A period of five years from the date of the search was defined, considering studies from 2019 to 2024, to obtain the most recent evidence on the subject. This time limit was defined because, during the primary research, it was found a literature review [27] and a scoping review [31] on authentic leadership as of 2018, so it was decided to examine authentic leadership in nursing from that date, starting the research in 2019. The database search was conducted in September 2024.

### 2.4. Paper Selection

In paper selection process, only articles with the full text available and that met the defined eligibility criteria were considered. The selection of studies was conducted in three sequential stages: analysis of titles, reading of abstracts and, whenever necessary, reading of the full texts to clarify eligibility. Only studies that fully met the previously defined criteria were included in the review.

The search strategy initially resulted in a total of 2457 records, whose references were exported from the databases and subsequently imported into the Rayyan^®^ platform (version 1.6.0) to facilitate the management and screening of studies. Using this tool, duplicates (253) were removed, and the articles were organised according to the previously defined eligibility criteria.

The screening of studies was carried out independently by two reviewers in Rayyan^®^ (blind mode), and a third reviewer was involved to analyze the articles and resolve discrepancies. After eliminating duplicates, 2204 records remained for analysis. Of these, 12 were selected based on a review of the titles and abstracts. This was followed by a full reading of the texts, which resulted in the inclusion of 10 studies, as they fully met the eligibility criteria.

In addition, the reference lists of the included studies were analysed to identify additional potentially relevant publications, resulting in the inclusion of four articles. Two studies were also added based on recommendations from experts in the field.

At the end of the process, 16 studies were considered eligible and included in this review. Figure 1 shows the results of the article analysis stages, according to the PRISMA Flow Diagram [33].

### 2.5. Data Extraction

To extract data from the articles included in the review, a results extraction table (Table 1) was used.

Data extraction was performed independently by three reviewers using a table developed specifically for this purpose, in accordance with the research question and previously defined objectives, as recommended by the JBI methodology [32]. The table structure included, where applicable, the following elements: Author(s), Year of publication and Country; Title; Objectives; Study design; Study population (including sample size and participants); Context; Data collection instrument(s); Main results. It was not necessary to contact the authors of the included studies for clarification or to obtain additional data, as the available information allowed the objectives of the review to be met.

### 2.6. Data Synthesis

The data synthesis followed a meta-aggregation approach as proposed by Lockwood et al. (2015) [34]. This method involves the identification of findings from included studies, grouping similar findings into categories and generating synthesised statements to reflect overarching themes. Seven thematic categories were generated through an inductive process based on the convergence of findings across studies.

## 3. Results

After evaluating all articles for eligibility according to the defined inclusion criteria, 16 articles were included in this scoping review. These were published between 2019 and 2024. This review includes responses from 10,753 nurses working in a hospital context.

The articles originated in different countries, including: Brazil (12.5%; *n* = 2) [2,18]; Egypt (12.5%; *n* = 2) [8,36]; United States of America (12.5%; *n* = 2) [6,42]; Jordan (12.5%; *n* = 2) [9,28]; Oman (12.5%; *n* = 2) [37,40]; Turkey (12.5%; *n* = 2) [11,39]; Saudi Arabia (6.25%; n = 1) [1]; China (6.25%; *n* = 1) [5]; South Korea (6.25%; *n* = 1) [38]; and Portugal (6.25%; *n* = 1) [41].

Geographical analysis showed that the Asian continent had the highest number of publications (68.8%; *n* = 11) [1,5,8,9,11,28,36,37,38,39,40]. This was followed by the American continent (25%; *n* = 4) [2,6,18,42]. Europe was represented by only one study (6.25%; *n* = 1), conducted in Portugal [41].

Regarding the methodological design of the included studies, there was a predominance of quantitative investigations of a cross-sectional, descriptive and/or correlational nature (81.3%; *n* = 13) [1,2,5,9,11,18,28,36,37,38,40,41,42]. In addition to these, two quasi-experimental studies were identified (*n* = 2; 12.5%) [8,39]. A study employing an exploratory and confirmatory factor analysis approach was also included (6.25%; *n* = 1) [6].

We used meta-aggregation according to Lockwood et al. [34]. Through the analysis of the themes and correlations established by the studies, seven categories were found: effects of authentic leadership on nurses’ well-being [5,9,11,36,37,41]; relationship between authentic leadership and performance/professional behaviours [1,5,9,11,18,41]; influence of authentic leadership on job satisfaction, organisational commitment and intention to stay [6,18,28,38]; perception and knowledge about authentic leadership (leader versus followers) [2,8,18]; relationship between authentic leadership and the nursing practice environment [37,40,41,42]; authentic leadership training [8,39]; influence of sociodemographic and cultural factors on authentic leadership [28,38].

Regarding data collection instruments, all articles used more than one instrument, some of which were used repeatedly but translated and validated for the specific cultural context, allowing correlations to be established between the same variables. A sociodemographic questionnaire was used in all studies to characterise the sample.

Eleven articles used the Authentic Leadership Questionnaire (ALQ) [1,2,5,6,9,18,28,37,38,39,40]. Two articles used the Authentic Leadership Scale [11,41]. Two articles used the Authentic Nurse Leadership Questionnaire (ANLQ) [6,36]. One article used the Authentic Leadership Self-Assessment Questionnaire [35]. One article used the Kessler Psychological Distress Scale [36]. One article used the Maslach Burnout Inventory [37]. Two articles used the Practice Environment Scale of the Nursing Work Index (PES-NWI) [37,40]. One article used the Turnover Intention Scale [9]. One article used the Six Dimensions of Nurse’s Performance Scale [9]. One article used the First-line nurse managers’ self-assessment of Authentic Leadership behaviour questionnaire [8]. One article used the Nursing care self-efficacy questionnaire [8]. One article used the Nurses’ trust in their workplace questionnaire [8]. One article used the Utrecht Work Engagement Scale (UWES) [28]. One article used the Organizational Commitment Questionnaire [38]. One article used the Job Satisfaction Survey [18]. One article used the Safety Climate Survey (SCS) [39]. One article used the Conditions of Work Effectiveness Questionnaire-II (CWEQ-II) [39]. One article used the Psychological Empowerment Instrument (PEI) [39]. One article used the Leadership Self-Efficacy Scale (LSES) [40]. One article used the Motivation to Lead Scale (MLS) [40]. One article used a sociodemographic questionnaire with questions about leadership developed by the study researchers [2]. One article used The Scale for the Intention to Quit [11]. One article used the Employee Performance Scale [11]. One article used the Critical Elements of a Healthy Work Environment Scale [42]. One article used the Nurse Well-Being Index [42]. One article used the Psychological Capital Questionnaire [5]. One article used the Caring Behaviour Inventory [5].

In some articles [1,8,36,38,41,42], it was not clear which data collection instrument was used; only the purpose of the instrument was clarified.

## 4. Discussion

The objective of this scoping review was to examine the characteristics of authentic leadership among nurses in a hospital context based on scientific evidence. Analysis of the 16 selected studies reveals a growing diversity of scientific evidence in this area of knowledge, which supports the importance of authentic leadership in hospital nursing practice, with an emphasis on its positive effects on professionals and the work environment.

### 4.1. Effects of Authentic Leadership on Nurses’ Well-Being

Authentic leadership has a direct and indirect impact on nurses’ well-being and mental health, reducing burnout and the intention to leave [9,36,37]. These studies highlight the importance of authentic leaders as protective factors against emotional exhaustion. The dimension of “Self-awareness” emerges as particularly relevant, being associated with lower burnout [37].

Furthermore, the study by Zhang et al. [5] identified psychological capital as a mediator of the relationship between authentic leadership and caring behaviour, indicating that the authenticity of leaders strengthens the emotional well-being of professionals, which, in turn, favours the quality of care provision. The study by Ozer et al. [11] also suggests that authentic leadership, by promoting a decision-making process based on the “Balanced Processing” dimension, contributes to healthier work environments with greater emotional support.

However, Marques-Quinteiro et al. [41] point out that the association between authentic leadership and mental health indicators such as flourishing (being well and functioning well) may depend on organisational conditions. This reinforces the importance of favourable work contexts to maximise the benefits of authentic leadership.

### 4.2. Relationship Between Authentic Leadership and Performance/Professional Behaviours

Authentic leadership has been positively associated with individual and collective performance [1,9,18,41], as well as professional behaviours aligned with excellence in care [5]. Thus, the practice of authentic leadership by leader’s benefits care provision and influences the behaviours of the nurses they lead.

The study by Ozer et al. [11] demonstrated that professional performance benefits from behaviours associated with the “Balanced Processing” dimension, which involves considering different perspectives prior to decision-making, favouring fairer and contextually adjusted decisions.

### 4.3. Influence of Authentic Leadership on Job Satisfaction, Organisational Commitment and Intention to Stay

Studies by Assi et al. [28], Baek et al. [38], Batista et al. [18] and Giordano-Mulligan and Eckardt [6] show that authentic leadership is associated with higher levels of job satisfaction and organisational commitment. These findings reinforce the strategic relevance of authentic leadership, especially in hospital contexts, where organisational challenges require resilient, motivated teams that are aligned with institutional values. These variables are crucial for the stability of nursing teams and the quality of care.

The study by Assi et al. [28] revealed a positive and statistically significant correlation between organisational commitment and the “Self-awareness” dimension of authentic leadership. This dimension refers to the leader’s ability to recognise their own emotions, values, strengths and limitations, critically reflecting on their impact on relationships and decisions [6]. Thus, the results obtained by Assi et al. [28] suggest that self-aware leaders are perceived as having integrity, transparency and consistency, characteristics that can foster a sense of belonging to the organisation, increasing organisational commitment.

Regarding the intention to stay, the study by Abousoliman and Hamed [36] showed that authentic leadership is inversely related to nurse turnover. In other words, the greater the perception of authenticity in leaders—expressed through transparency in relationships, ethical consistency and genuine involvement with teams—the less likely nurses are to leave the organisation. This finding enhances the central role of authentic leadership in retaining talent, as it is a protective factor against turnover.

### 4.4. Perception and Knowledge About Authentic Leadership (Leader Versus Followers)

It is important to emphasise the differences between leaders’ and their subordinates’ perceptions of authentic leadership [18]. The results found in the study by Batista et al. [18] indicate that leaders’ authentic leadership scores are higher than those perceived by their subordinates in all dimensions of the Authentic Leadership Questionnaire instrument. This discrepancy may result from a self-worth bias among leaders, different understandings of authentic leadership constructs, or limited opportunities for subordinates to consistently observe authentic behaviours in practice. This discrepancy indicates the need to strengthen communication and continuous feedback, as well as to implement programmes that encourage self-reflection among managers, such as those developed in the studies by El-Fattah Mohamed Aly et al. [8] and Dirik and Intepeler [39], which promote leadership that is more aligned with the expectations and needs of teams.

Moreover, Mondini et al. [2] highlight that knowledge about authentic leadership is still limited among nurses, which reinforces the importance of awareness and continuous training for this current of thought on leadership, which may be the future solution to ensure the effectiveness and sustainability of hospital contexts.

### 4.5. Relationship Between Authentic Leadership and the Nursing Practice Environment

Another relevant point to highlight is the influence of authentic leadership in building a healthy work environment. Several studies point to authentic leadership as a determining factor in strengthening the nursing practice environment, promoting favourable conditions for both quality care and the well-being of professionals [41,42].

This association emphasises the importance of training and valuing authentic leaders who can create work environments based on transparency, ethics and trust. The studies by Al Sabei et al. [37], Labrague et al. [40], Marques-Quinteiro et al. [39] and Raso et al. [42] are consistent in this regard, demonstrating that authentic leaders favour healthy work environments.

It is important to note that, according to Marques-Quinteiro et al. [41], authentic leadership is positively related to team performance, but only when working conditions are favourable. This result reinforces that the effectiveness of authentic leadership may depend on organisational support and existing contextual conditions, underlining the importance of environments that enable the full exercise of this current of leadership thought.

### 4.6. Authentic Leadership Training

The evidence presented by El-Fattah Mohamed Aly et al. [8] and Dirik and Intepeler [39] clearly reinforces the effectiveness of authentic leadership training programmes in the nursing context. Both studies, using a quasi-experimental methodology, demonstrated that structured training interventions have a significant impact on managers’ leadership skills, with positive effects on the individual, the organisation and the quality of care [8,39].

In the study by Dirik and Intepeler [39], there was a significant increase in authentic leadership scores, both among leaders and their employees, after the intervention. In addition, there was an improvement in the safety environment, structural empowerment and psychological empowerment of professionals [39]. Authentic leadership and structural empowerment emerged as significant predictors of patient safety, underscoring the importance of authentic leaders in creating safer work environments [39]. Among the dimensions of authentic leadership, “Relational Transparency” and “Self-Awareness” showed significant improvements after the training programme, highlighting them as key aspects in training programmes [39].

The study by El-Fattah Mohamed Aly et al. [8] revealed statistically significant improvements in managers’ knowledge of authentic leadership, as well as in their self-assessment as leaders, throughout and after the implementation of the programme. These advances were reflected in higher levels of self-efficacy in nursing care and greater confidence among nurses in their clinical practice [8]. The authors also demonstrated that the knowledge acquired during the training had a direct effect on nurses’ self-assessment, with consistent improvement across all dimensions of authentic leadership [8].

These results support the notion that training programmes focused on authentic leadership can be highly effective in developing more conscious, consistent and effective leaders in hospital contexts. In terms of implications for practice, the data discussed emphasise the importance of continuous training policies and sustained institutional programmes geared towards developing authentic skills in nurse managers. These programmes not only enhance the professional growth of leaders, but also promote safer and more trusting environments, with a direct impact on the quality of care provision.

### 4.7. Influence of Sociodemographic and Cultural Factors on Authentic Leadership

International analysis of studies indicates that the application and impact of authentic leadership can be influenced by cultural and socio-geographical context. Studies from countries with different healthcare systems and cultures, such as Brazil, China, Egypt, Jordan, Oman, South Korea, Turkey, Portugal and the United States of America, point to nuances in the perception and expression of authentic leadership, although its importance is universally recognised.

As for demographic factors such as gender, age and professional experience, the articles analysed indicate that they can influence the perception and manifestation of authentic leadership [9,28,38]. These variables should be considered in future studies and in the implementation of leadership development programmes, so that they are tailored to the specific needs of nurses in different contexts and with different personal and professional characteristics.

The predominance of cross-sectional studies limits the ability to infer cause-and-effect relationships, enhancing the need for longitudinal or experimental research to deepen understanding of the temporal and causal impact of authentic leadership. The few quasi-experimental studies included, such as El-Fattah Mohamed Aly et al. [8] and Dirik and Intepeler [39], reinforce the effectiveness of training programmes aimed at authentic leadership. The absence of qualitative or mixed-method studies demonstrates a gap in the literature regarding the understanding of nurses’ subjective perceptions and experiences in the context of authentic leadership.

Finally, even though there is consensus about the benefits of authentic leadership, the present methodological limitations should be considered for future studies. Thus, new research should favour longitudinal or qualitative designs, which can deepen the perceptions, experiences and trajectories of nurses and leaders, expanding the knowledge and applicability of authentic leadership concepts in the hospital context.

Thus, it can be concluded that authentic leadership is important in nursing because it promotes the well-being of professionals, enhances performance as well as the quality and safety of nursing care, increases satisfaction and organizational commitment, and contributes to supportive work environments. It can be interpreted as both a protective factor and an institutional strengthening mechanism, whose impact depends on favorable organizational and cultural conditions, as well as on training programs aimed at developing authentic leadership competencies among nurses.

## 5. Limitations

Despite the relevant contributions of this review to the understanding of authentic leadership in the nursing context, it is important to recognise some limitations that may have influenced the results obtained.

Firstly, although a systematic and comprehensive research strategy was carried out, and despite searching six databases, it is possible that not all relevant evidence was identified.

In addition, the inclusion of studies published only in languages of greater expression in the international scientific literature, namely Portuguese, English and Spanish, may have limited the scope of the review. The exclusion of studies written in other languages was due to reasons related to the linguistic domain of the reviewers, which constitutes a methodological limitation.

Furthermore, a significant proportion of the studies included in this review were conducted in different geographical, cultural and economic contexts, which may compromise the comparison and transferability of results, especially in countries with fewer resources or different health and education systems. The effectiveness and perception of authentic leadership behaviours are partly shaped by cultural factors, which makes it difficult to define a universally effective leadership profile in culturally diverse teams.

Another aspect to consider is the scarcity of longitudinal studies analysing the sustained impact of authentic leadership over time, both in terms of organisational performance and the continuous professional development of nurses. Most of the available data focuses on short-term effects, so further analysis of medium- and long-term results is needed. In addition, the scarcity of qualitative studies is also a limitation, as it does not allow for an understanding of the subjective perceptions of both leaders and employees. Therefore, future studies should use longitudinal and qualitative designs that allow for the expansion of the theoretical basis of authentic leadership.

## 6. Implications for Practice

The results of this review clearly point to the relevance of authentic leadership as a strategic factor in promoting healthy, safe and motivating work environments in the hospital context. The evidence gathered supports the notion that authentic leaders contribute significantly to the well-being of nurses, team performance, the quality of care provided and organisational sustainability.

In this sense, one of the main practical implications is the need to invest in authentic leadership development and training programmes, focusing on the dimensions of authentic leadership. Studies by El-Fattah Mohamed Aly et al. [8] and Dirik and Intepeler [39] have shown that well-structured training interventions promote significant improvements in professionals’ self-efficacy, confidence and empowerment, with a direct impact on the quality of care and patient safety. Additionally, these programmes enhance both leaders’ and team members’ perceptions of authenticity in leadership.

Healthcare institutions should therefore integrate these training programmes into their organisational development policies, with the aim of training self-aware, ethical leaders who excel in transparency and authenticity, capable of responding effectively to the complex challenges of hospital environments. Such initiatives contribute to the retention of professionals and the strengthening of organisational commitment.

Another relevant aspect concerns the importance of aligning leaders’ perceptions with those of their subordinates in relation to authentic behaviours. The discrepancy identified by Batista et al. [18] between leaders’ self-assessment and their teams’ perceptions highlight the need to promote continuous feedback, self-reflection and formative supervision processes. These mechanisms favour a more realistic and functional alignment of expectations.

In addition, it is essential to consider the socio-cultural and demographic specificities of the contexts and professionals involved. Cultural diversity and differences in generation, gender or professional experience influence the perception and expression of authentic leadership. Therefore, it is recommended that training programmes and implementation strategies are adapted to the realities of each work context and the characteristics of the teams, respecting diversity as an enriching factor.

Finally, the data analysed reinforces the idea that the effectiveness of authentic leadership depends not only on the leader’s skills but also on organisational conditions. The study by Marques-Quinteiro et al. [41] emphasised that the positive impact of authentic leadership on team performance occurs when the work environment is favourable. Thus, it is essential that institutions create structural and organisational conditions that enable the effective practice of authentic leadership, namely through institutional support, participatory management and policies that promote professional well-being.

## 7. Conclusions

This scoping review allowed us to synthesise and critically analyse the available scientific evidence on authentic leadership among nurses in a hospital context. The results show that authentic leadership represents a promising leadership approach for the effective management of nursing teams, as it promotes healthy work environments, improves performance and well-being of nurses, with a consequent increase in the quality of care and the safety of professionals and patients.

Positive correlations were found between authentic leadership and variables, such as organisational commitment, job satisfaction, empowerment, performance, motivation to lead and lower levels of burnout and turnover intention. These findings confirm its multidimensional and beneficial impact across individuals and organisations levels.

Specific dimensions of authentic leadership stood out, namely “Self-awareness”, “Moral and Ethical Perspective” and “Transparency” (according to the Authentic Leadership Questionnaire), as structural elements for building trusting relationships and strengthening the nursing practice environment.

Training programmes aimed at developing authentic leadership were found to be effective in improving leadership behaviour fostering greater alignment between leaders and their teams, and enhancing outcomes such as self-efficacy, confidence and practice safety. These programmes are therefore recommended as institutional strategies for promoting leadership grounded in ethical values and relational transparency.

Despite significant contributions, the review has some limitations. First, the inclusion of studies only in English, Spanish, and Portuguese may have restricted the scope of findings. Additionally, most included studies were cross-sectional, limiting the possibility of establishing causal relationships. The scarcity of qualitative or longitudinal research also constrains understanding of the deeper mechanisms and temporal effects of authentic leadership. So, it is recommended that longitudinal, qualitative research be developed to deepen understanding of the mechanisms through which authentic leadership influences nursing outcomes. Sociocultural, demographic and contextual considerations should also be integrated into future approaches to ensure the applicability and effectiveness of authentic leadership in different healthcare settings.

In summary, authentic leadership proves to be an essential strategic resource for the sustainability of healthcare organisations, promoting the well-being, trust and commitment of nurses, which are fundamental pillars for excellence in hospital care and healthcare system success.

## Figures and Tables

**Figure 1 healthcare-13-02713-f001:**
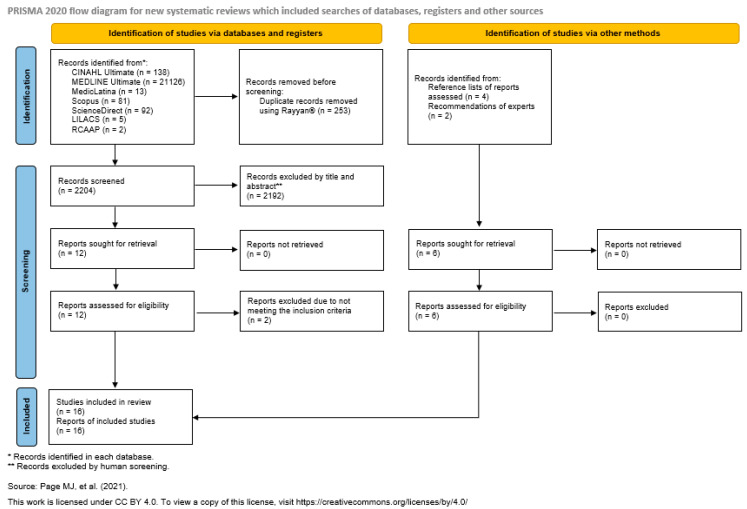
PRISMA Flow Diagram [35] representing the paper selection process.

**Table 1 healthcare-13-02713-t001:** Characteristics of the studies analysed.

Author(s) Year of Publication Country	Title	Study Design	Sample Size/ Perception	Data Collection Instrument(s)	Main Results
Abousoliman & Hamed, 2024 [36] Egypt	Effect of authentic leadership on Nurses’ psychological distress and turnover intention	Quantitative Descriptive Cross-sectional	285 nurses from one hospital Perception of the Subordinates	- Authentic leadership self-assessment questionnaire - Kessler psychological distress scale - Three-item scale to measure intention to leave	Authentic leadership is significantly negatively associated with psychological distress. A statistically significant positive relationship was found between psychological distress and intention to leave. Authentic leadership has a positive impact on nurses and is inversely associated with turnover.
Al Sabei et al., 2023 [37] Oman	The influence of nurse managers’ authentic leadership style and work environment characteristics on job burnout among emergency nurses	Quantitative Cross-sectional	160 emergency nurses from six hospitals Perception of the Subordinates	- Maslach Burnout Inventory - Authentic Leadership Questionnaire (ALQ) - Practice Environment Scale of the Nursing Work Index	More than two-thirds of nurses reported high levels of burnout. Authentic leadership and a supportive work environment were significantly associated with lower burnout. The ‘Self-Awareness’ dimension scored the highest, while the ‘Balanced Processing’ dimension scored the lowest. Transparency and moral and ethical conduct were also negatively associated with burnout. A positive correlation was found between authentic leadership and the team nurses’ perception of nursing practice environment. A significant negative correlation was found between nursing practice environment and burnout. The moral and ethical conduct of nurse leaders is essential to support nurses in making ethical decisions.
Alilyyani, 2022 [1] Saudi Arabia	The Effect of Authentic Leadership on Nurses’ Trust in Managers and Job Performance: A Cross-Sectional Study	Quantitative Cross-sectional	116 nurses from one hospital Perception of the Subordinates	- Authentic Leadership Questionnaire (ALQ) - Data collection instrument for measuring confidence in managers - Scale for assessing nurses’ performance	Authentic leadership influences employee performance and behaviour by enhancing trust in both managers and nurses. No significant relationship was found between authentic leadership and job performance. Significant and positive relationships were identified between authentic leadership and trust in managers.
Allan & Rayan, 2023 [9] Jordan	Association Between Authentic Leadership in Nurse Managers and Performance and Intention to Leave Among Registered Nurses	Quantitative Cross-sectional Correlational	166 nurses from one hospital Perception of the Subordinates	- Sociodemographic questionnaire - Authentic Leadership Questionnaire (ALQ) - Turnover Intention Scale - Six Dimensions of Nurse’s Performance Scale	A positive correlation was found between authentic leadership and team performance, and a statistically significant negative correlation was found between authentic leadership and the intention to leave. Nurses who described their managers as autocratic reported higher turnover intention than those who described their managers as having democratic or laissez-faire leadership styles. Regardless of the nurses’ sociodemographic status, authentic leadership is associated with better performance and lower turnover intention.
Assi et al., 2024 [28] Jordan	Nurse Managers’ Authentic Leadership and their Relationship with Work Engagement among Registered Nurses	Quantitative Cross-sectional Correlational	238 nurses Perception of the Subordinates	- Authentic Leadership Questionnaire (ALQ) - Utrecht Work Engagement Scale (UWES)	A statistically significant positive correlation was found between authentic leadership and commitment to work. The “Self-awareness” dimension of authentic leadership had the strongest correlation with commitment to work. The “Moral and Ethical Perspective” dimension obtained the highest score, while the “Transparency” dimension obtained the lowest score. Authentic leadership revealed significant differences in gender, indicating that male participants had significantly higher scores. Authentic leadership was found to be significantly and positively correlated with participants’ age, nursing experience and experience at their current hospital.
Baek et al., 2019 [38] South Korea	Authentic leadership, job satisfaction and organizational commitment: The moderating effect of nurse tenure	Quantitative Cross-sectional	1118 nurses from six hospitals Perception of the Subordinates	- Authentic Leadership Questionnaire (ALQ) - One item scale assessing workplace satisfaction - Organizational Commitment Questionnaire	Authentic leadership as perceived by nurses was positively correlated with job satisfaction and organisational commitment. The strength of these correlations was attenuated with nurse tenure and was no longer significant for nurse managers with more than 20 years of service. Hospitals should provide leadership programmes to nurture authenticity in their workers. Managers should recognise that their authenticity can positively affect their team and should adopt an ethical and honest attitude at work. To strengthen the importance of authentic leadership among nursing managers, they should be evaluated not only on performance, as has traditionally been done, but also on their relational traits.
Batista et al., 2021 [18] Brazil	Authentic leadership, nurse satisfaction at work and hospital accreditation: Study in a private hospital network	Quantitative Cross-sectional	282 nurses (94 leaders and 188 subordinates) from 11 hospitals Perception of the subordinates and the leaders	- Authentic Leadership Questionnaire (ALQ) - Job Satisfaction Survey	A significant difference was found between the assessment of leaders and followers in all dimensions of the ALQ. Regarding the association of authentic leadership with job satisfaction, a significantly positive correlation was found among followers. As for leaders, no significant correlation was identified between job satisfaction and the dimensions of authentic leadership. The results indicate that leaders’ authentic leadership scores are higher than those perceived by their subordinates in all dimensions of the ALQ instrument. The domain of job satisfaction that had the lowest score was “Operational Conditions”. By demonstrating authentic leadership behaviours, leaders increase nurses’ satisfaction and performance.
Dirik & Intepeler, 2024 [39] Turkey	An authentic leadership training programme to increase nurse empowerment and patient safety: A quasi-experimental study	Quasi-experimental	36 nurse managers and 153 nurses Perception of the subordinates and the leaders	- Safety Climate Survey (SCS) - Authentic Leadership Questionnaire (ALQ) - Conditions of Work Effectiveness Questionnaire-II (CWEQ-II) - Psychological Empowerment Instrument (PEI)	After the intervention, the safety environment and the authentic leadership scores increased among both leaders and employees. Structural and psychological empowerment also increased among followers. Authentic leadership and structural empowerment were found to be predictors of the safety environment. Healthcare organisations should implement programmes focused on authentic leadership and nurse empowerment to increase patient safety. Psychological empowerment was not predictive of the patient’s safety. Regarding the dimensions of authentic leadership, a significant difference was found in the dimensions of “Relational Transparency” and “Self-Awareness”.
El-Fattah Mohamed Aly, 2023 [8] Egypt	Effectiveness of first-line nurse manager authentic leadership training program on nurses’ attitudes in medical and surgical care units	Quasi experimental	36 nursing managers and 300 nurses from medical and surgical wards Perception of the subordinates and the leaders	- Questionnaire assessing authentic leadership administered to managers - First-line nurse managers’ self-assessment of authentic leadership behaviour questionnaire - Nursing care self-efficacy questionnaire - Nurses’ trust in their workplace questionnaire	A statistically significant improvement was observed in both the knowledge of nursing managers and the self-assessment of authentic leadership, which led to improved self-efficacy in nursing care and nurses’ confidence in nursing practice after the implementation of the training programme. Managers’ knowledge of authentic leadership throughout the programme had a positive correlation and a direct effect on nurses’ self-assessment. Managers’ self-assessment of authentic leadership behaviours throughout and after the programme predicted positive self-efficacy in nursing care and nurses’ confidence. The improvement in managers’ self-assessment was higher in all dimensions of authentic leadership behaviour throughout the programmes and after them, compared to the pre-training programme.
Giordano-Mulligan & Eckardt, 2019 [6] United States of America	Authentic Nurse Leadership Conceptual Framework Nurses’ Perception of Authentic Nurse Leader Attributes	Exploratory and confirmatory factor analysis	309 nurses Perception of the subordinates	- Authentic Nurse Leadership Questionnaire (ANLQ) - Authentic Leadership Questionnaire (ALQ)	The ANLQ is valid and reliable; the concepts of the ANLQ were statistically supported by exploratory and confirmatory factor analyses. The ANLQ better identified nursing values (compared to the ALQ) through a stronger relationship with professional life and organisational commitment. Data emerged to support the newly developed conceptual framework and the five subscales, in which “Caring” was the new dimension identified. The relationship between work and personal life and authentic leadership in nursing supported the ANLQ. The relationship between authentic leadership in nursing and commitment to nursing demonstrated that the ANLQ had a positive correlation with commitment. The importance of leadership in nursing for the provision of high-quality healthcare is unquestionable. Nursing leaders who embody attributes of the conceptual framework of authentic leadership can promote a healthy professional life and greater commitment to the nursing profession.
Labrague et al., 2021 [40] Oman	Authentic leadership and nurses’ motivation to engage in leadership roles: The mediating effects of nurse work environment and leadership self-efficacy	Quantitative Cross-sectional	1534 nurses from 24 intensive care units Perception of the Subordinates	- Practice Environment Scale of the Nursing Work Index (PES-NWI) - Authentic Leadership Questionnaire (ALQ) - Leadership Self-Efficacy Scale (LSES) - Motivation to Lead Scale (MLS)	Authentic leadership has a significant and positive relationship with nurses’ motivation to engage in formal leadership roles. The nursing practice environment and leadership self-efficacy partially mediated the association between authentic leadership and nurses’ motivation to engage in formal leadership roles. Authentic leadership was positively and moderately correlated with leadership self-efficacy. Authentic leadership had a significant direct effect on the nursing practice environment. The authenticity of nurse managers influences nurses’ perceptions of their nursing practice environment.
Marques-Quinteiro et al., 2021 [41] Portugal	On the Relationship Between Authentic Leadership, Flourishing, and Performance in Healthcare Teams: A Job Demands-Resources Perspective	Quantitative Cross-sectional	106 nurses from two hospitals Perception of the Subordinates	- Authentic Leadership Scale - Scale to measure Flourishing - Scale to measure team performance	Authentic leadership is positively related to team performance, but only when working conditions are favourable. No positive relationship was found between authentic leadership and flourishing (defined as the combination of feeling good and functioning well, with a high level of mental flourishing symbolising mental health). Authentic leadership promotes positive work environments. Team members have fewer emotional reactions to stressful factors and, therefore, are in a better psychological state when leaders provide clear goals, a clear clarification of roles and strategies for better performance.
Mondini et al., 2020 [2] Brazil	Authentic leadership among nursing professionals: Knowledge and profile	Quantitative Cross-sectional	84 nurses from one hospital Perception of the Subordinates	- Sociodemographic questionnaire with questions about leadership - Authentic Leadership Questionnaire (ALQ)	The nurses were unfamiliar with authentic leadership but scored high on authentic leadership behaviours. The nurses identified communication, planning and organisation as leadership skills. The nursing team identified transparency, morale, ethics and self-awareness as domains of authentic leadership. Holding a leadership position and keeping up to date had a positive influence on authentic leadership behaviours. Working individually had a negative influence on authentic leadership behaviours.
Ozer et al., 2019 [11] Turkey	The Relationship Between Authentic Leadership, Performance and Intention to Quit the Job of Nurses	Quantitative Cross-sectional	189 nurses from one hospital Perception of the Subordinates	- Authentic Leadership Scale - The Scale for the Intention to Quit - Employee Performance Scale	The relationships found between the dimensions of authentic leadership and intention to leave are low and statistically insignificant. The correlations between performance and the dimensions of authentic leadership are low and significant. An increase in behaviours associated with the dimension of “balanced processing” increases employee performance.
Raso et al., 2022 [42] United States of America	Perceptions of nurses and leaders on authentic nurse leadership, healthy work environment, intent to leave and nurse well-being during a second pandemic year	Quantitative Cross-sectional Descriptive Correlational	1795 nurses and nursing leaders from around 3000 hospitals Perception of the subordinates and the leaders	- Authentic Nurse Leadership Questionnaire - Critical Elements of a Healthy Work Environment Scale - Intention to leave (2 questions) - Nurse Well-Being Index	Authentic leadership among nurses correlated positively with a healthy work environment, highlighting the importance of leadership for the nursing practice environment. Authentic leadership in nursing, a healthy work environment and the well-being of nurses are essential components of efforts to stabilise the nursing workforce. Nurses perceived a high level of authentic leadership in their managers. A significant negative correlation was found between a healthy work environment and nurse well-being and authentic leadership and nurse well-being, indicating that nurses are not satisfied with their nursing practice environment and consider themselves to be unwell.
Zhang et al., 2023 [5] China	The mediating role of psychological capital on the relationship between authentic leadership and nurses’ caring behaviour: a cross-sectional study	Quantitative Cross-sectional	3662 nurses from 37 hospitals Perception of the Subordinates	- Authentic Leadership Questionnaire (ALQ) - Psychological Capital Questionnaire - Caring Behaviour Inventory	Psychological capacity significantly mediated the relationship between authentic leadership and nurses’ care. Positive and significant correlations were found between authentic leadership, psychological capital and caring behaviour. Authentic leadership is strongly associated with nurses’ caring behaviour, with psychological capital as a mediator. Nursing managers can activate nurses’ caring behaviour by improving nurses’ psychological capital and the effectiveness of authentic leadership.

## Data Availability

To obtain the data supporting the results presented, please contact the authors of this scoping review.

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
