# Peer review of "Authentic Leadership of Nurses in the Hospital: A Scoping Review"

_healthcare, 2025, doi:10.3390/healthcare13212713_

Round 1
Reviewer 1 Report (Previous Reviewer 1)
Comments and Suggestions for Authors
After carefully reviewing the revised version of the manuscript and the detailed responses to each observation, I have no further comments.
The authors have adequately and comprehensively addressed all the reviewers’ recommendations, including clarification of inclusion criteria and grey literature handling, expansion of the discussion with strengths, limitations and a forward research agenda, as well as thorough style and reference corrections.
Author Response
Please see the attachment.

Reviewer 2 Report (Previous Reviewer 3)
Comments and Suggestions for Authors
Thank you for your efforts to review this manuscript. The paper is significantly improved from the previous version. However, the following comment has not been addressed satiafactorily:
3. Results and Discussion
Much more detail is needed in the results section. It is not sufficient to just list the themes identified- these need to be discussed and described here. Some of the information in the discussion section is actually results, as it is a description of the literature related to the identified themes. Please move this from the discussion to the results section. Your discussion then needs to take a higher level view of the topic, drawing out key insights and relating these to the existing literature and current nursing practice.
I therefore feel the paper needs further refinements before it is suitable for publication.
Author Response
Please see the attachment

Reviewer 3 Report (New Reviewer)
Comments and Suggestions for Authors
The manuscript is a timely and much-needed piece of work. It addresses the important question of how authentic leadership by nurse leaders impacts professionals and the organisation in the hospital setting, a highly demanding environment.
In our opinion, the authors should make some changes to the document in order for it to be accepted.
Our suggestions are as follows:
-
Line 37 refers to the selection of 14 cross-sectional studies and 2 quasi-experimental studies, making a total of 16 selected studies, which is confirmed in the attached flowchart. However, lines 407-411 when discussing the methodological analysis, it is stated that there are 13 cross-sectional studies, 2 quasi-experimental studies, and 1 study with another methodology that is not clearly specified (specific methodological approach). In this case, we would appreciate it if the results provided were standardised and the methodology followed by the study described were explained in more detail.
-
Most of the selected articles are cross-sectional and quantitative studies, which is a limitation described by the authors in the discussion (lines 816-823). As these are cross-sectional studies, it is difficult to establish the causality of the relationships highlighted in the article, despite proposing solutions in the limitations section.
-
In line 168, the abbreviation NPE is used. It would be important to define the meaning of the abbreviation beforehand to facilitate the reader's understanding.
-
In line 562, there is an error, as it says ‘Tabel’ instead of ‘table’. In this table, it would be useful to add a column differentiating whether the study measured the perception of the leader, the perception of the subordinate, or both, as this would facilitate a quick understanding of the source of the results.
-
The article explains that there is a perceptual discrepancy between leaders and subordinates, but it does not address the reason for these discrepancies in depth. It would be very beneficial to explore this discrepancy.
Author Response
Please see the attachment

This manuscript is a resubmission of an earlier submission. The following is a list of the peer review reports and author responses from that submission.
Round 1
Reviewer 1 Report
Comments and Suggestions for Authors
I appreciate the opportunity to review this manuscript. The topic of authentic leadership in hospital nursing is timely and very relevant, especially with the need for leadership strategies that promote professional well-being, patient safety, and organizational sustainability. The paper is well-written and provides a comprehensive overview of the evidence, although some methodological and structural improvements could enhance its impact and clarity.
This is a scoping review conducted in line with JBI and PRISMA-ScR guidance, with the protocol registered on OSF. It synthesizes evidence from 16 eligible articles (2019–2024), covering more than 10,000 nurses internationally.
Novelty (few comparable reviews since 2018), protocol registration, and a well-structured synthesis by thematic categories.
Limited critical appraisal of evidence strength, underdeveloped discussion of heterogeneity and potential biases, and lack of comparative synthesis tools (tables, figures, conceptual models).
Comments by section
Title and Abstract
The abstract clearly states the objectives, databases, time frame, and main findings, but it lacks detail on inclusion/exclusion criteria, handling of grey literature, and the strength of evidence.
Introduction
The objective is stated but could be sharpened by specifying the expected outputs (e.g., identification of knowledge gaps, a research agenda).
Please close with a sentence that clearly defines the study’s purpose and contribution.
Methods
Further detail is needed on inter-reviewer reliability (e.g., kappa or agreement), the inclusion of grey literature, and whether any critical appraisal tools were used.
Please expand the methods to include these elements.
Results
The thematic organization is strong (well-being, performance, job satisfaction, perceptions, environment, training, sociodemographic factors). A PRISMA flow diagram and a descriptive table of studies are included.
However, comparative tables and visual synthesis are missing (e.g., instruments used, geographic distribution, most relevant outcomes).
Please add summary tables, for example:
-
Table 1: Study-level summary (author/year, country/setting, design, sample size, instrument used, key outcomes, main findings).
-
Table 2: Outcome-by-outcome map (outcome, number of studies, direction/consistency of effects, indicative strength of evidence).
-
Table 3 (optional): Instruments and metrics used across studies (constructs, reliability, cut-offs).
Also consider a conceptual figure showing the pathways by which authentic leadership may influence nurse and patient outcomes.
Discussion
Positive findings are emphasized without sufficiently qualifying the strength of evidence or potential biases.
Please add a subsection on Strengths and limitations of the evidence, discuss heterogeneity and likely sources of bias, and propose a forward research agenda (longitudinal studies, qualitative work, intervention trials).
References
There are some duplications and formatting inconsistencies (e.g., “[5][5]”).
Kindly review style carefully, remove duplicates, and ensure DOIs are provided throughout.
Minor recommendations (optional)
Page 2 (Abstract)
- Lines 23-24: “keep professionals motivated and satisfied, and meet the organisation's objectives.” Change to: “keep professionals motivated and satisfied while meeting the organisation’s objectives” for smoother flow.
- Lines 39-40: “Conclusion: Authentic leaders have a positive impact on the quality of nursing care, patient health outcomes, professional satisfaction and motivation to lead and the achievement of healthcare institution goals.” Change to: split into two sentences for clarity: “…motivation to lead. They also contribute to the achievement of healthcare institution goals.”
Page 4 (Introduction)
- Lines 65-66: “It was also found a literature review dated 2018 on authentic leadership…”
Change to: “A literature review dated 2018 on authentic leadership was also found…” (more natural word order).
Page 5 (Methods)
- Lines 73-74: “The eligibility criteria were defined based on the PCC mnemonic (participants, concept, context), in accordance with the methodology proposed by the Joanna Briggs Institute (JBI). [25].” Change to: remove the extra full stop before the reference: “…Institute (JBI) [25].”
Page 7 (Results)
- Lines 210-211: “The majority of the included studies were cross-sectional (n = 14), with only two quasi-experimental studies.” Change to: “Most included studies were cross-sectional (n = 14); only two were quasi-experimental.”
Page 13 (Discussion)
- Lines 425-427: “This highlights the need to strengthen authentic leadership interventions in training and in organisational contexts.” Change to: “This highlights the need to strengthen authentic leadership interventions in both training and organisational contexts.”
Page 18 (Acknowledgments and Funding)
- Line 572: “The APC was founded by Healthcare.” Correction: should read “The APC was funded by Healthcare.”
Pages 20–22 (References)
- Several references show typographical duplication (e.g., “[5][5]”). Please review carefully and remove duplicates.
- Ensure consistent journal abbreviations and add missing DOIs.
Reviewer 2 Report
Comments and Suggestions for Authors
It appears that the authors may have uploaded an incorrect version of the manuscript for submission. A finalized version is required for a more thorough review. The first round of review has been conducted based on this incomplete file.
The manuscript contains unresolved field code errors (e.g., ‘Código de campo alterado’, references), which indicates it is not in a finalized format. Please revise and upload the manuscript according to the journal’s submission guidelines and format requirements.
1. Introduction
‘The section beginning with ‘Authentic leadership is a complex concept…’ (line 76) is somewhat repetitive and vague. I recommend summarizing the core attributes of authentic leadership more concisely—for example, self-awareness, relational transparency, and ethical standards—so that readers can quickly grasp its essential characteristics.
The justification for the study in the last paragraph (lines 129–139), beginning with ‘We decided to develop a scoping review…’, is somewhat underdeveloped. While it mentions a knowledge gap, it would benefit from a clearer articulation of the limitations of previous reviews and the distinct contribution of the present study. For example, you could emphasize that earlier reviews were not specifically focused on nursing and did not adequately incorporate the most recent research, which would strengthen the rationale for conducting this scoping review.
Some sentences are excessively long, which reduces readability (e.g., lines 95–104). I recommend breaking them into shorter sentences to improve clarity and logical flow.
2. Materials and Methods
The manuscript states that the review was limited to studies in English, Spanish, and Portuguese. To strengthen the methodological rigor, please provide a brief justification for excluding studies in other languages, such as feasibility or resource constraints.
Although meta-aggregation is mentioned as the synthesis method, the description of how data were analyzed and integrated remains insufficient. Please clarify whether the synthesis involved only tabular presentation or whether studies were thematically grouped and aggregated according to JBI guidance.
It is noted that three reviewers independently screened the studies and resolved discrepancies by consensus. To enhance transparency, I recommend elaborating on the consensus process—for example, whether it involved structured discussion, predefined rules, or the involvement of a third reviewer in case of disagreement.
3. Results
Please organize the Results section according to the journal guidelines
4. Discussion ~
Throughout the Discussion section, several arguments (e.g., the association of authentic leadership with job satisfaction, reduced burnout, and healthy work environments) are repeated multiple times. I recommend condensing these points into a concise synthesis to highlight the key messages and avoid unnecessary redundancy.
I recommend using a reference management tool to ensure consistency and accuracy in the citation style, as there are several duplicated or incorrectly formatted references throughout the manuscript.
The Discussion section is overly descriptive and overlaps with the Results. I recommend separating the two more clearly: ‘what was found’ should remain in the Results, while the Discussion should focus on ‘why it matters’ and ‘how it can be interpreted.’
The Discussion emphasizes the positive effects of authentic leadership but provides limited critical appraisal. I suggest expanding the discussion of methodological limitations (e.g., predominance of cross-sectional studies, cultural/contextual influences) and addressing potential biases and limits to generalizability.
The Conclusion section is lengthy and repetitive. Please streamline it in line with the journal’s guidelines, focusing on (i) a concise summary of the main findings, (ii) practical implications, and (iii) limitations and recommendations for future research.
Reviewer 3 Report
Comments and Suggestions for Authors
Thank you for the opportunity to review this paper. It is an interesting paper and makes an important contribution to the field.
I offer the following comments for your consideration:
1. Introduction
The introduction provides a thorough overview of existing research in this area and makes a strong case for the present study. However, it would be useful to more thoroughly link this to nursing practice and standards, e.g. do any nursing codes, guidelines or standards of practice emphasise the importance of leadership or transformational leadership to nursing practice? Also, you mention there are a number of current thoughts in leadership and authentic leadership is one of these- please provide a brief summary of others to better orient the reader to this area.
2. Methods
Methods are generally well-described. However, please describe your method of data synthesis (i.e. meta aggregation) here.
3. Results and Discussion
Much more detail is needed in the results section. It is not sufficient to just list the themes identified- these need to be discussed and described here. Some of the information in the discussion section is actually results, as it is a description of the literature related to the identified themes. Please move this from the discussion to the results section. Your discussion then needs to take a higher level view of the topic, drawing out key insights and relating these to the existing literature and current nursing practice.
4. References
The manuscript is supported by appropriate references and your referencing style is sound.
5. Other comments
Please provide a copy of the completed PRISMA-ScR checklist in the supplementary material.
Thank you and I look forward to reading the next version of the paper!